# Evidence of Self-Association and Conformational Change in Nisin Antimicrobial Polypeptide Solutions: A Combined Raman and Ultrasonic Relaxation Spectroscopic and Theoretical Study

**DOI:** 10.3390/antibiotics12020221

**Published:** 2023-01-20

**Authors:** Afrodite Tryfon, Panagiota Siafarika, Constantine Kouderis, Spyridon Kaziannis, Soghomon Boghosian, Angelos G. Kalampounias

**Affiliations:** 1Department of Chemistry, University of Ioannina, GR-45110 Ioannina, Greece; 2Department of Physics, University of Ioannina, GR-45110 Ioannina, Greece; 3Department of Chemical Engineering, University of Patras, GR-26504 Patras, Greece; 4University Research Center of Ioannina (URCI), Institute of Materials Science and Computing, GR-45110 Ioannina, Greece

**Keywords:** Nisin, polypeptides, Raman, ultrasonic relaxation spectroscopy, self-association

## Abstract

The polypeptide Nisin is characterized by antibacterial properties, making it a compound with many applications, mainly in the food industry. As a result, a deeper understanding of its behaviour, especially after its dissolution in water, is of the utmost importance. This could be possible through the study of aqueous solutions of Nisin by combining vibrational and acoustic spectroscopic techniques. The velocity and attenuation of ultrasonic waves propagating in aqueous solutions of the polypeptide Nisin were measured as a function of concentration and temperature. The computational investigation of the molecular docking between Nisin monomeric units revealed the formation of dimeric units. The main chemical changes occurring in Nisin structure in the aqueous environment were tracked using Raman spectroscopy, and special spectral markers were used to establish the underlying structural mechanism. Spectral changes evidenced the presence of the dimerization reaction between Nisin monomeric species. The UV/Vis absorption spectra were dominated by the presence of π → π* transitions in the peptide bonds attributed to secondary structural elements such as α-helix, β-sheets and random coils. The analysis of the acoustic spectra revealed that the processes primarily responsible for the observed chemical relaxations are probably the conformational change between possible conformers of Nisin and its self-aggregation mechanism, namely, the dimerization reaction. The activation enthalpy and the enthalpy difference between the two isomeric forms were estimated to be equal to *ΔH*_1_* = 0.354 ± 0.028 kcal/mol and *ΔH*_1_^0^ = 3.008 ± 0.367 kcal/mol, respectively. The corresponding thermodynamic parameters of the self-aggregation mechanism were found to be *ΔH*_2_* = 0.261 ± 0.004 kcal/mol and *ΔH*_2_^0^ = 3.340 ± 0.364 kcal/mol. The effect of frequency on the excess sound absorption of Nisin solutions enabled us to estimate the rate constants of the self-aggregation mechanism and evaluate the isentropic and isothermal volume changes associated with the relaxation processes occurring in this system. The results are discussed in relation to theoretical and experimental findings.

## 1. Introduction

In recent years, infection management, resistance to antibiotics and infectious disease control have attracted intensive scientific research. Furthermore, the development of new compounds with essential antibacterial properties in order to control food spoilage bacteria is of crucial importance to anti-spoilage food packaging, thus leading to safer food supply. The World Health Organization, aiming to prevent antibiotic resistance, recognises the development of new antibacterial compounds and, more specifically, antibiotic peptides as promising materials for this purpose [1]. Direct use of antibiotic peptides or combined use of these materials with other antimicrobials facilitates the development of new strategies to control bacterial growth [2]. 

Nisin is a water-soluble and non-toxic polypeptide that is composed of 34 amino acid compounds and is a representative material with various applications in the food industry nowadays, assigned to E234. Nisin belongs to lantibiotics and is commercially developed using the Gram-positive bacterium *Lactococcus lactis* on natural substrates, such as milk or dextrose. In the food industry, Nisin is usually obtained from the culturing of *L. lactis*, and it is not chemically synthesized. Subtilin and Epidermin are related to Nisin and are also members of the lantibiotic class. This system acts effectively against *Staphylococci*, *Lysteria* and *Clostridia* [3]. Nisin acts effectively at levels nearing the parts-per-billion range. It is mainly used in processed cheese, meats, beverages, etc., at levels ranging from ~1 to 25 ppm, depending on the food type and regulatory approval [4]. The concentration of Nisin can be measured using several techniques, including chromatographic methods, simple agar diffusion bioassays, etc. [4]. A recent application of Nisin is in food packaging applications, where it can serve as a preservative via controlled release onto the food surface from the polymer packaging [5].

The structure of an isolated Nisin molecule is conformationally stabilized by five thioether macrocyclic rings. It is important to achieve progress on the comprehensive understanding of Nisin behaviour at the molecular level in the solution state, where several structural and dynamic processes may take place, such as self-association, cooperative conformational transitions and protolytic reactions. The combined use of vibrational and acoustic relaxation spectroscopy and theoretical calculations provides important insights into microstructural interactions and dynamical phenomena in the liquid state [6,7,8]. Furthermore, the results of molecular docking studies can be used as a starting model of the molecular interactions among ligand and receptor molecules. 

In our study towards the understanding of the molecular relaxation dynamics of aqueous Nisin solutions, we employed ultrasonic relaxation spectroscopy to obtain temperature- and concentration-dependent acoustic spectra. The systematic analysis of the acoustic spectra allowed us to explore the relaxation processes and disentangle processes with similar time scales. A comprehensive model is proposed that accounts for both relaxation processes observed experimentally in the studied system. Specific thermodynamic parameters, such as the potential barrier of the conformational change and self-association reaction, as well as the associated standard volume change, as functions of Nisin concentration in the solutions were evaluated and discussed. As part of this work, Raman and ultrasound absorption measurements in Nisin solutions were also performed to establish the possibility of a self-association scheme. Molecular docking and density functional theory (DFT) computational methodologies were utilized to theoretically examine the possibility of self-association and estimate the properties of the isolated Nisin molecule and aggregates. Moreover, the associated volume change due to the self-association of Nisin was estimated and compared with the experimental findings.

## 2. Results and Discussion

### 2.1. Dynamic and Structural Processes in Aqueous Nisin Solutions

The molecular structure of Nisin after applying the optimization procedure is presented in Figure 1. Two distinct molecular structures of the polypeptide Nisin exist in nature, the so-called Nisin A and Nisin Z. The structure of the Nisin A molecule is presented in Figure 1. It is composed of 34 amino acids with amino and carboxyl end-groups and five internal ring structures that involve disulphide bridges. The polypeptide Nisin possesses three relatively unusual amino acids. These are dehydroalanine, lanthionine and β-methyllanthionine amino acids. Furthermore, the amino acid Lanthionine is common in several bacteriocins, known as lantibiotics. The difference between Nisin Z and Nisin A is the substitution of asparagine with histidine in the polypeptide molecular structure. Even though Nisin Z has antimicrobial activity comparable to that of Nisin A, Nisin Z exhibits greater diffusion ability in agar gels. 

The Nisin molecule is dominated by the presence of amide, amine, carboxylic and hydroxyl functional groups and is involved in structural mechanisms, such as conformational changes, molecular self-aggregation reactions and proton-transfer reactions. The conformational change between possible conformers is described as
(1)$$Nisin⇆{\left(Nisin\right)}^{*}$$
where (*Nisin*)* represents the possible conformer. Considering the possible stepwise self-aggregation mechanism as a mean reaction, we can write
(2)$$nNisin\;⇆{\left(Nisin\right)}_{n}$$
where *n* is the relatively low aggregation number and (*Nisin*)*_n_* represents the aggregate. This mechanism takes place in moderately dilute solutions, while at higher concentrations, possible solute–solvent interactions may appear. The proton-transfer reaction may be described as
(3)$$Nisin+H_{2}O\;⇆NisinH^{+}\ldots OH^{-}⇆NisinH^{+}+OH^{-}$$

### 2.2. Elucidation and Assignment of the Observed Structural Mechanisms

On the left side of Figure 2a, the structure of the Nisin molecule is presented after the optimization procedure that is described in the experimental section. After performing the molecular docking study, we obtained the Nisin dimer structure that is presented on the right side of Figure 2a. More complicated structures, such as tetramers, were not stable enough, as revealed by the theoretical calculations. So, the proposed self-aggregation reaction of Nisin is presented in Figure 2a with aggregation number *n* = 2. The possibility of formation of dimeric and tetrameric structures of Nisin has been reported in the literature [9].

To determine if this mechanism is indeed observed experimentally during the dilution of Nisin, we calculated the theoretically predicted Raman spectra corresponding to the binding regions of the two monomers constituting the dimer denoted by ellipses in Figure 2b. Since it is not feasible to perform quantum mechanical calculations covering the full dimer structure, we only isolated the specific narrow parts of the structure that are observed in Figure 2c,d. 

In parallel, we recorded the Raman spectra of all solutions in the dilute region corresponding to concentrations from 0 (solvent) to 5 mM in 1 mM increments. The spectroscopic results are shown in Figure 3a. The Raman spectrum of crystalline Nisin is also presented for comparison. It seems that the solution spectra above 3000 cm^−1^ are dominated by the strong bands attributed to hydroxyl groups of the solvent and Nisin. The N-H stretching band is also observed near ~3270 cm^−1^, characteristic of the amide group. At lower frequencies, the peaks located near ~1656 and ~1537 cm^−1^ are assigned to Amide I and Amide II, respectively, and are both related to peptide linkages. To evaluate the presence of the dimer, we directly compared the experimental spectra of solid Nisin and of the 5 mM solution with the theoretically predicted spectra corresponding to the binding regions of the two monomers constituting the dimer (ellipses in Figure 2b), as shown in Figure 3b. The direct comparison revealed that the theoretical spectra coincide significantly with the experimental Raman spectra even though the calculation performed was only a rough estimation in the vapor state. The peaks observed in the theoretically calculated vibrational spectra for the two parts of the dimer are also present in the experimental spectrum of the aqueous solution of Nisin, mainly in the frequency range of 1500–1700 cm^−1^.

The UV absorption spectra of Nisin solutions were examined between 190 and 700 nm and are shown in Figure 4. The absorption spectra in the range of 180 to 230 nm are dominated by the presence of π → π^*^ transitions in the peptide bonds [10]. More specifically, the spectral region from 190 to 210 nm is attributed to secondary structural elements such as α-helix, β-sheets and random coils. The 210–230 and 270–280 bands are characteristic absorbance patterns of most peptide antibiotics [10]. The monotonous variation in the absorbance patterns in the low-wavelength region of the spectra is indicative of the gradual aggregation process occurring in the studied system. 

### 2.3. Effects of Concentration and Temperature on the Relaxation Behaviour

Ultrasonic relaxation spectra were obtained in a Nisin concentration range of 1 to 10 mM in 0.02 M HCl (aqueous) solvent and are shown in Figure 5. For all solutions, the absorption and velocity measurements in the pure solvent revealed the absence of any relaxational effects in the frequency range covered in this work. The total ultrasonic absorption coefficient per squared frequency, namely, the ratio of *a*/*f*^2^, has two distinct contributions. The first one is the classical, (*a*/*f*^2^)_*classical*_, or non-relaxing part, and the second one is the excessive, (*a*/*f*^2^)_*excessive*_, or relaxing part [11,12,13]. Classical absorption *a* is analogous to *f*^2^, and it is common practice to plot the absorption data in the form of *α*/*f*^2^ as a function of frequency, since this representation is a straight line parallel to the frequency axis. Thus, any additional relaxation is clearly identified as an excess. The classical, (*a*/*f*^2^)_*classical*_, or non-relaxing term is the sum of four terms attributed to vibrational relaxation, viscous absorption, thermal absorption and radiation. In many cases, the experimental relaxing absorption coefficient, (*a*/*f*^2^)_experimental_, follows Debye-type frequency dependence [11,12,13].
(4)$${\left(\frac{a}{f^{2}}\right)}_{experimental}=\sum \frac{A_{i}}{1+{\left(\frac{f}{f_{r_{i}}}\right)}^{2}}+{\left(\frac{a}{f^{2}}\right)}_{classical}=\sum \frac{A_{i}}{1+{\left(\frac{f}{f_{r_{i}}}\right)}^{2}}+B$$
where subscript *i* represents the *i*-th process of the system with the corresponding $f_{r_{i}}$ relaxation frequency and *A_i_* is the amplitude of the relaxation. The contribution of classical absorption derived from vibrational relaxation, visco-thermal absorption and radiation discussed above is represented in Equation (4) with constant *B* for simplicity and is considered frequency independent. 

In our case, the experimental data could not be fitted adequately with Equation (4). The data were best fitted using the real part of the Cole–Davidson distribution function [14,15].
(5)$${\left(\frac{a}{f^{2}}\right)}_{experimental}=\sum \left\{A_{i}{\left(cos\varphi_{i}\right)}^{\beta_{i}}cos\left(\beta_{i}\varphi_{i}\right)\right\}+B$$
where *φ_i_* = tan^−1^(*ωτ*) with *ω* = 2*πf* and *τ_i_ =* 2*π/f_ri_* is the corresponding relaxation time. Parameter *β_i_* is related to the width of the distribution of the *i*-th process and lies between 0 and 1. When *β_i_* equals 1, then the distribution function leads to the Debye form. Generally, parameter *β_i_* permits the adequate description of different spectral shapes that are more stretched for *β_i_* < 1 and extend over a wider range of frequencies on a logarithmic scale than the simple Debye relaxation. 

A representative example of the fitting procedure used to describe the ultrasonic absorption data as a function of frequency in Nisin solutions at 20 °C is presented in Figure 6. It is evident that the Cole–Davidson distribution function fit adequately the experimental *a*/*f*^2^ values. As already discussed, the acoustic spectra of polypeptides may exhibit three main mechanisms, namely, the conformational changes between possible conformers, the self-aggregation mechanism taking place in the moderately dilute region and the proton-transfer reaction expected to occur at higher concentrations. These mechanisms were also expected to be present in the spectra of Nisin solutions. Nevertheless, we only observed two distinct relaxation mechanisms in the acoustic spectra in the relatively low MHz region of frequency. The first process observed at lower frequencies is attributed to conformational changes, while the second one observed at higher frequencies is assigned to the self-aggregation of Nisin. Indeed, conformational change is expected to occur at lower energies than the self-aggregation mechanism [11]. 

The fitting procedure was performed by minimizing the following ratio:(6)$$\frac{{\left\{{\left(\frac{a}{f^{2}}\right)}_{experimental}-{\left(\frac{a}{f^{2}}\right)}_{calculated}\right\}}^{2}}{{\left(\frac{a}{f^{2}}\right)}_{experimental}}$$
which allowed us to estimate the numerous free fitting parameters. Among these fitting parameters, the characteristic relaxation frequency (*f_r_*) and the relaxation amplitude (*A*) are useful since they enable the quantitative characterization of the relaxation effects to be performed. The concentration dependence of the characteristic ultrasonic relaxation frequency (*f_ri_*) and relaxation amplitude (*A_i_*) with *i* = 1 and 2 at 20 °C for both mechanisms is shown in Figure 7. Subscript 1 denotes the acoustic parameters related to the conformational changes, while subscript 2 represents the acoustic parameters associated with the self-aggregation of Nisin. Relaxation amplitude *A*_1_ exhibited a monotonous increasing trend with the increase in solution concentration, while relaxation frequency *f_r_*_1_ appeared to be concentration independent. This behaviour is typical of relaxation associated with unimolecular equilibrium between different rotational isomers [11,12,13]. It seems that thermal relaxation plays a predominant role in Nisin solutions. Relaxation amplitude *A*_2_ and characteristic frequency *f_r_*_2_ of the second relaxation process observed at higher frequencies both exhibited a monotonous increasing trend with the increase in Nisin concentration in solutions, supporting our proposed assignment of this ultrasonic relaxation to the self-aggregation mechanism. The classical contribution to absorption coefficient *B* appeared to be almost concentration independent in the frequency range covered here. 

Analogous results were obtained for the experimental absorption coefficients and the corresponding fitting curves as a function of temperature for a Nisin solution with C = 3 mM. The acoustic absorption spectra as functions of frequency at all temperatures studied are presented in Figure 8. 

In addition, in the temperature-dependent measurements, the Cole–Davidson distribution function adequately fitted the experimental *a*/*f*^2^ values, and the so-obtained fitting parameters are presented in Figure 9 for both relaxation processes. Both relaxation amplitudes *A*_1_ and *A*_2_ increased with the temperature, while the corresponding relaxation frequencies, *f_r_*_1_ and *f_r_*_2_, exhibited a clear linear blue shift with the temperature. 

The activation enthalpies and the enthalpy differences associated with the conformational change and self-aggregation mechanism can be assessed using the temperature dependence of the corresponding fitting parameters. From the slope of the Arrhenius-type diagram of ln(*f_r_/T*) versus 1*/T* shown in Figure 10a,c for low- and high-frequency relaxation, we could estimate the activation enthalpy for the unimolecular equilibrium between different rotational isomers and dimerization reaction of Nisin, respectively. Indeed, based on the Eyring theory, the characteristic relaxation frequency, *f_r_*, of a relaxation process is associated with the corresponding thermodynamic parameters and the following equation holds [13,16,17,18,19]:(7)$$\frac{1}{2\pi f_{r}}=\frac{h}{k_{B}T}\exp\left(\frac{\Delta H^{*}-T\Delta S^{*}}{R}\right)$$
or
(8)$$ln\left(\frac{f_{r}}{T}\right)=-\frac{\Delta H^{*}}{1000R}\left(\frac{1000}{T}\right)+\left[\frac{\Delta S^{*}}{R}+ln\left(\frac{k_{B}}{2\pi h}\right)\right]$$
where *ΔH** is the activation enthalpy and *ΔS** is the activation entropy. All the other symbols have their usual meanings, i.e., *k_B_* is the Boltzmann constant, while *h* and *R* are the Plank constant and the gas constant, respectively. Subsequently, we can determine enthalpy difference *ΔH*^0^ with the following equation [13,19]:(9)$$\frac{T\mu_{max}}{u^{2}}=\frac{\pi \rho V\Theta^{2}{\left(\Delta H^{0}\right)}^{2}}{2JC_{P}^{2}}exp\left(\frac{\Delta S^{0}}{R}\right)\;exp\left(-\frac{\Delta H^{0}}{RT}\right)$$
where *μ_max_* is directly estimated using the relaxation parameters of *f_r_* and *A* and the sound speed is estimated as [13,19]
(10)$$\mu_{max}=\frac{1}{2}A\;u\;f_{r}$$

In Equation (9), parameters *V*, *Θ* and *C_P_* represent the molar volume, the thermal expansion coefficient, and the specific heat at constant pressure, respectively. The mass density of the solution is designated as *ρ*, while *J* = 4.187 J/cal is the conversion factor from Joule to calories. Bearing in mind that the temperature range of the experiments was relatively constricted, we can treat the enthalpy difference as being temperature independent and estimate this thermodynamic parameter using the slope of the semi-log plot of (*Tμ_max_*/*u^2^*) vs. 1/*T* presented in Figure 10b,d. Equation (9) is valid when the criterion of *r* << 1 is fulfilled [20]. Quantity *r* represents the relaxation strength and is given by [20]
(11)$$r_{i}=\frac{A_{i}u_{i}f_{r_{i}}}{\pi }-\frac{1}{4}\left(1-\frac{2B_{i}}{A_{i}}\right){\left(\frac{A_{i}u_{i}f_{r_{i}}}{\pi }\right)}^{2}$$

By means of the above equation, we estimated parameters *r*_1_ and *r*_2_, which correspond to the low-frequency and high-frequency relaxation attributed to conformational change and aggregation mechanism, respectively. The relaxation strength, the solution density and the related acoustic parameters are summarized in Table 1 for a Nisin solution at concentration C = 3 mM at all temperatures studied, and in Table 2, the results are shown considering all concentrations at 20 °C. Indeed, the values of *r*_1_ and *r*_2_ for both mechanisms are by far lower than unity; thus, Equation (9) provides accurate results. 

All the plots presented in Figure 10 exhibit a clear linear dependency, confirming the initial hypotheses that the activation enthalpies and the enthalpy differences are temperature independent. For the conformational change, the activation enthalpy and the enthalpy difference were estimated to be equal to *ΔH*_1_*** = 0.354 ± 0.028 kcal/mol and *ΔH*_1_^0^ = 3.008 ± 0.367 kcal/mol, respectively. The corresponding thermodynamic parameters for the aggregation of Nisin were valued to be equal to *ΔH*_2_*** = 0.261 ± 0.004 kcal/mol and *ΔH*_2_^0^ = 3.340 ± 0.264 kcal/mol.

### 2.4. Kinetic Study of the Self-Aggregation Reaction of Nisin

Considering that the self-aggregation reaction of Nisin is presented as one mean stage (Equation (2)) and that the process, kinetically, is in the ultrasonic time domain, then relaxation frequency *f_r_*_2_ and reactant concentration *C* are associated by the following equation [21,22]: (12)$$2\pi f_{r_{2}}=n^{2}k_{f}C^{n-1}+k_{b}$$
where parameters *k_f_* and *k_b_* represent the forward and backward rate constants, respectively; and parameter *n* denotes the aggregation number and is always an integer. For several integer values of *n*, we plotted the product 2*πf_r_*_2_ against *C^n−^*^1^, which was expected to be linear, and subsequently estimated the corresponding Pearson coefficient. The best Pearson coefficient of R = 0.98378 was obtained for *n* = 2, and we estimated the forward and backward rate constants from the slope and intercept of the plot to be equal to *k_f_* = 5.98 × 10^7^ M^−1^s^−1^ and *k_b_* = 1.25 × 10^7^ s^−1^, respectively. 

### 2.5. Volume Changes Accompanying the Relaxation Processes

By applying a sound wave to the aqueous solution of Nisin, the system was perturbed from equilibrium, causing a finite volume change. The calculation of the volume change magnitude can be performed directly using the associated acoustic parameters. More specifically, isentropic volume change $\Delta V_{s_{i}}$ of each process can be determined with [23]
(13)$$\mu_{max_{i}}=\frac{\pi \rho u^{2}\Gamma_{i}}{2RT}{\left(\Delta V_{s_{i}}\right)}^{2}$$
where subscript 1 denotes the maximum value of the absorption per wavelength at relaxation frequency *f_r_*_1_ related to the conformational changes, while subscript 2 denotes the same parameter associated with the self-aggregation mechanism of Nisin. Furthermore, *Γ_i_* is the concentration parameter associated with the progress of the *i*-th process. The concentration parameter can be calculated with the following equation [24]:(14)$$\Gamma =\frac{1}{V}{\left[\frac{\partial }{\partial lnK_{C}}\left(\frac{n_{j}-n_{j}^{0}}{\nu_{j}}\right)\right]}_{p,T}$$
where *ν_j_* are the stoichiometric coefficients of the *j*-th species involved in the chemical reaction and *n_j_^0^* denotes the mole number in the equilibrium state. 

Isothermal volume change $\Delta V_{T_{i}}$ can be estimated using the isentropic volume change with the following equation [23]:(15)$$\Delta V_{T_{i}}=\Delta V_{s_{i}}+\frac{a_{p}\Delta H_{i}}{\rho C_{P}}$$

In Equation (15), the thermal expansivity of solution *α_p_* and heat capacity under constant pressure *C_p_* were assumed to be equal to the corresponding solvent properties since our most concentrated solution was 0.010 M Nisin. The ratio of $\frac{a_{p}\Delta H}{\rho C_{P}}$ represents the enthalpic contribution to the isentropic volume change and is relatively small compared with the isothermal and isentropic volume change terms. The so-obtained isothermal and isentropic volume changes associated with the conformational change and aggregation mechanism are presented in Figure 11a,b, respectively. Indeed, the ratio of $\frac{a_{p}\Delta H}{\rho C_{P}}$ is modest compared with the isothermal volume change due to both relaxation processes taking place in Nisin solutions. 

As can be seen in Figure 11, it seems that the volume change due to the aggregation mechanism was strongly affected by the solution concentration. On the contrary, the volume change associated with the conformational change was almost insensitive to the concentration variation. The calculated volume change between rotational isomers is expected to be lower than the volume change associated with a mechanism, such as the self-aggregation mechanism, that has a much stronger impact on the structure. The stronger the effect on the structure is, the larger the volume change is. 

Theoretically, we can estimate the standard volume change, *ΔV*_2_, associated with the self-aggregation reaction of Nisin as
(16)$$\Delta V_{2}={\bar{V}}_{{\left(Nisin\right)}_{n}}-n{\bar{V}}_{Nisin}$$
where ${\bar{V}}_{{\left(Nisin\right)}_{n}}$ represents the partial molar volume of aggregate (*Nisin*)*_n_*, ${\bar{V}}_{Nisin}$ denotes the partial molar volume of the monomeric Nisin species and aggregation number *n* equals 2. The optimized structure of the aggregate was obtained with the molecular docking study. The theoretical standard volume change of the aggregate (dimer) and monomer of Nisin was obtained by means of the PM6 semi-empirical method. Following this procedure, the standard volume change due to the self-aggregation of Nisin was found to be *ΔV*_2_ = 165.69 cm^3^/mol, which is comparable to the experimental findings in the dilute region, considering that the calculation was carried out in a vacuum environment. The volume change, *ΔV*_1_, associated with the conformational change of Nisin cannot be calculated theoretically due to the stochastic character of the conformation. Polypeptides usually exist as rod-like helical structures, rigid or even random coil forms. The transition among these forms and/or possible isomerization in each one of these forms may be affected by temperature, the solvent type, pHhhH, etc. Due to the inherent complexity of the Nisin molecular structure, it is not possible to handle theoretically possible conformational changes utilizing quantum mechanical approaches, and peculiar molecular dynamics methodologies should be adopted. 

### 2.6. Intermolecular Free Length of Nisin Solutions

Intermolecular free length *L* can be calculated directly using thermoacoustic parameters in an effort to study intermolecular interactions taking place in solutions. Free length is a simple intermolecular property describing the distance between the surfaces of the molecules. An empirical relation has been proposed to compute the intermolecular free length according to the following equation [25,26]:(17)$$L=K\sqrt{\beta_{s}}$$
where *K* is Jacobson’s constant, which is temperature and pressure dependent [27,28]. Furthermore, *β_s_* is the adiabatic compressibility of the system and is given by:(18)$$\beta_{s}=\frac{1}{\rho u^{2}}$$

The calculated values of intermolecular free length *L* as functions of solution concentration and temperature are presented in Figure 12a and Figure 12b, respectively. With the increase in the concentration of the solution at constant temperature and pressure, the intermolecular free length decreased monotonically (Figure 12a). The decrease was almost perfectly linear with Pearson R = −0.99676. This means that the distance between neighbouring molecules decreased with concentration, enhancing ultrasound wave propagation, which was reflected in the values of sound velocity (see Table 2). At higher concentrations, the molecular interactions were stronger, and molecular packing became rigid. 

On the other hand, with the increase in temperature, the molecular free length increased, following a rather polynomial trend under constant pressure and at specific solution concentrations (Figure 12b). A temperature rise resulted in an increase in the distance between the surfaces of the neighbouring molecules; intermolecular interactions were less pronounced; and molecular packing became looser. It seems that sound velocity and mass density have a direct relation with the intermolecular free length and may provide important information concerning intermolecular phenomena such as molecular association. 

## 3. Materials and Methods 

### 3.1. Materials

Crystalline solid Nisin from *Lactococcus lactis* 2.5% (balance sodium chloride) (Sigma-Aldrich, St. Louis, MO, USA) without any further purification was dissolved in HCl 0.02 M aqueous solution under ambient conditions. The concentrations of the solutions were 1, 2, 3, 4, 5, 7.5 and 10 mM. All prepared solutions were clear and light yellow in colour. The mass of solid used for each solution is presented in Table 3. The solvent was prepared from 37% HCl (Merck) and triply distilled water. The specific solvent was chosen, since Nisin exhibits greater solubility and stability in an acidic environment [29]. All solutions were subjected to measurements immediately after preparation.

### 3.2. Raman Spectroscopy

The 532 nm line of a continuous-wave diode laser (Spectra-Physics, Santa Clara, CA, USA) was used to excite Raman spectra in a 90° scattering geometry at the resolution of 2 cm^−1^. The scattered light was analysed using a single monochromator (IHR-320 JY; ISA-Horiba group, Kyoto, Japan) equipped with a CCD detector. In order to reject elastic Rayleigh scattering, a notch filter was used. The power of the laser on the sample was set below 70 mW to avoid heating the solution. Accumulation time of a few minutes was enough to achieve an adequate signal-to-noise ratio. A CCl_4_ reference sample was used to correct possible drift of the monochromator. Additional experimental details can be found elsewhere [30,31].

### 3.3. Ultrasonic Relaxation Spectroscopy 

A parallel-path pulse methodology was used to perform measurements of the ultrasonic absorption coefficient by employing sinusoidal burst in the desired frequency range. The repetition rate was set so as to avoid overlapping between the backscattered echoes and the subsequent pulse signal. To send and receive signals, two broad-band piezoelectric transducers were used with identical technical characteristics. The liquid sample was loaded in the temperature-controlled acoustic cell with a fixed optical path length of 1 cm. Common medical ultrasound gel was used as couplant between the transducers and the acoustic cell in an effort to establish perfect contact among surfaces and proper transmission of the generated pulse. The cell temperature was controlled within ±0.01 °C, and the accuracy of the sound absorption coefficient was better than ±5%. The ultrasound speed in the solutions was estimated by employing the pulse–echo overlap method with an experimental error of less than ±0.01%. Further information concerning the experimental procedures and additional details of the setup can be found in [32,33,34].

### 3.4. Computational Details

The structure of Nisin monomer (PDB ID: 1WCO) in electronic form (PDB file) was fetched from the RCSB database. Prior to molecular docking calculations, we optimized the obtained structure by utilizing the PM3 semi-empirical method. 

Computational investigation of the molecular docking between Nisin monomeric units was performed by means of AutoDock software (version 4.2). Initially, the size of the calculation box was set equal to 70 Å × 70 Å × 70 Å, large enough to adequately fit both Nisin molecules and make free docking possible. Since the Nisin molecule has a large number of rotating bonds, the backbone of the molecule was inputted as rigid, and the bonds of the side chains were left free to rotate. For all docking calculations, the assignment of partial charges was performed using Gasteiger charges [35,36,37]. The Lamarckian genetic algorithm (LGA) was employed for the pose selection, and at the end of the calculation, a free binding energy of −3.84 kcal/mol was found for the best pose. The sampling was extended, using 25,000 initial poses. The Lamarckian genetic algorithm was used to select the most probable pose.

After the molecular docking study, the so-obtained Nisin dimer structure was subsequently used to calculate vibrational frequencies and obtain the theoretical Raman spectra. Specifically, the binding regions of the two monomers were isolated, and quantum mechanical calculations were performed for the specific narrow parts of the structure (see Figure 2). It was not possible to perform quantum mechanical calculations covering the full dimer structure. The method chosen for the calculations belongs to the group of density functional theory (DFT) methods and is the hybrid method of Becke three-parameter exchange functional [38] with Lee–Yang–Parr correlation (B3LYP) [39]. The 6-31G(d) split valance basis set was used. The calculations were performed in a vacuum environment without potential interactions with solvent molecules [40]. 

For the theoretical calculation of standard volume change *ΔV_s_*, we used the PM6 semi-empirical method. After the calculation of the volume corresponding to Nisin monomer and dimer, standard volume change *ΔV_s_* was evaluated.

### 3.5. Additional Measurements

The electronic absorption spectrum was recorded using a UV-Vis (UV-1600PC) double-beam spectrophotometer (VWR) from 190 to 700 nm at 0.5 nm spectral resolution. 

Density measurements were performed using a temperature-controlled Anton Paar DMA cell with accuracy of ±0.0001 g/cm^3^. Dynamic (shear) viscosity was determined using the density and the kinematic viscosity values. The latter was measured by means of a Ubbelohde-type glass capillary viscosity meter (Schott, Mainz, Germany) at the desired temperature with accuracy of ±1%.

## 4. Conclusions

Ultrasonic absorption dispersion curves were measured as functions of concentration and temperature in aqueous Nisin solutions. Attenuation in excess of the pure solvent was found over the entire concentration and temperature ranges studied. The computational investigation of the molecular docking between Nisin monomers revealed the formation of dimeric units. Tetramer formation was not favoured based on theoretical calculations, although their existence has been reported in the literature. Raman spectroscopy was used to establish the underlying structural mechanism. The comparison between experimental and calculated Raman spectra by means of DFT methodologies evidenced the dimerization reaction between Nisin monomeric species. The UV/Vis absorption spectra established the presence of π → π* transitions in the peptide bonds, attributed to secondary structural elements such as α-helix, β-sheets and random coils. The systematic analysis of the acoustic spectra revealed the presence of two distinct relaxation mechanisms due to kinetics of the conformational conversion and self-aggregation of Nisin. The assumption that conformational change is the first mechanism was confirmed by the fact that the relaxation frequency did not change significantly in the studied concentration range. Raman and UV-Vis spectroscopy results support the existence of the self-aggregation phenomenon in nisin aqueous solutions. For the conformational change, the activation enthalpy and the enthalpy difference were found to be equal to *ΔH*_1_*** = 0.354 ± 0.028 kcal/mol and *ΔH*_1_^0^ = 3.008 ± 0.367 kcal/mol, respectively, while for the self-aggregation mechanism, they were estimated to be *ΔH*_2_*** = 0.261 ± 0.004 kcal/mol and *ΔH*_2_^0^ = 3.340 ± 0.364 kcal/mol. The theoretical standard volume change due to the self-aggregation of Nisin was found to be *ΔV*_2_ = 165.69 cm^3^/mol, which is comparable to the experimental value considering that the calculation was carried out in a vacuum environment. The volume change due to the aggregation mechanism was strongly affected by the solution concentration, while the volume change associated with the conformational change was almost insensitive to concentration variation. The volume change between rotational isomers was lower than the volume change associated with the dimerization mechanism, which had a much stronger impact on the structure. With the increase in temperature, intermolecular interactions were less pronounced, and molecular packing became looser. Concentration variation had the exact opposite effect. 

The obtained results are discussed in the framework of a proposed structural model highlighting the strong molecular association in this liquid system leading to complementary information on the structure and dynamics of Nisin. A comprehensive model of the whole relaxation processes is presented and discussed in detail. All of the above findings can be practically attributed to the alterations in the network rigidity due to variations in hydrogen bonding interactions at the molecular level. This study proves that the combination of Raman and ultrasonic relaxation spectroscopy, and theoretical calculations is a powerful diagnostic tool for the adequate evaluation of the processes occurring in polypeptide solutions. The elucidation of the detailed local structure is of key importance for the knowledge-based understanding of the antimicrobial and antibacterial performance. 

## Figures and Tables

**Figure 1 antibiotics-12-00221-f001:**
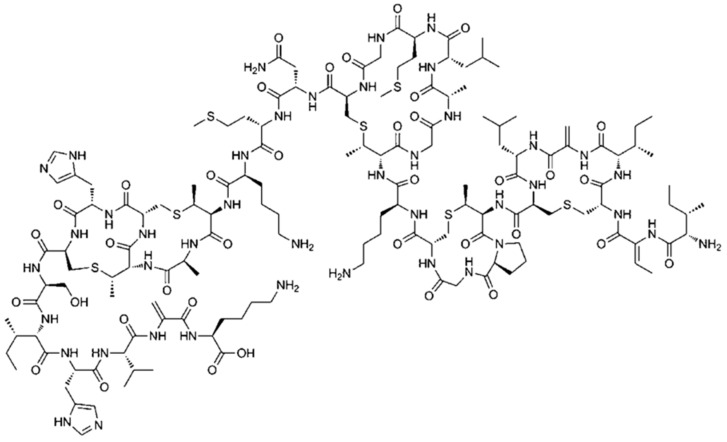
The molecular structure of polypeptide Nisin.

**Figure 2 antibiotics-12-00221-f002:**
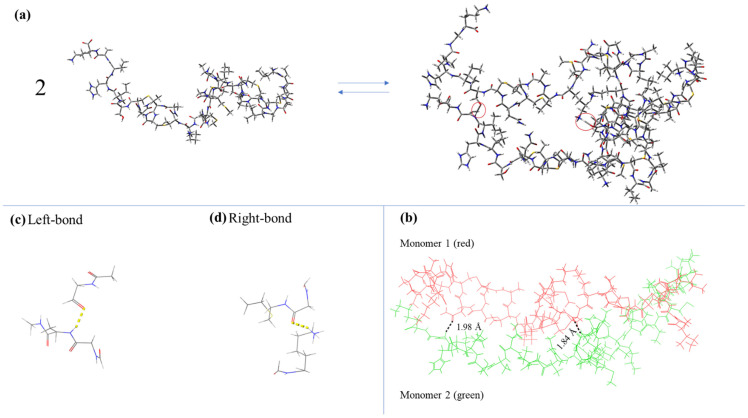
(**a**) Self-association reaction as predicted by the molecular docking study. Red circles denote the binding sites between Nisin monomers. (**b**) Binding regions of the two monomers constituting the dimer. (**c**,**d**) The two parts of the dimer structure of Nisin that were used for the theoretical calculation of vibrational frequencies denoted by ellipses. The yellow dashed line corresponds to a hydrogen bond between atoms belonging to a different monomer chain.

**Figure 3 antibiotics-12-00221-f003:**
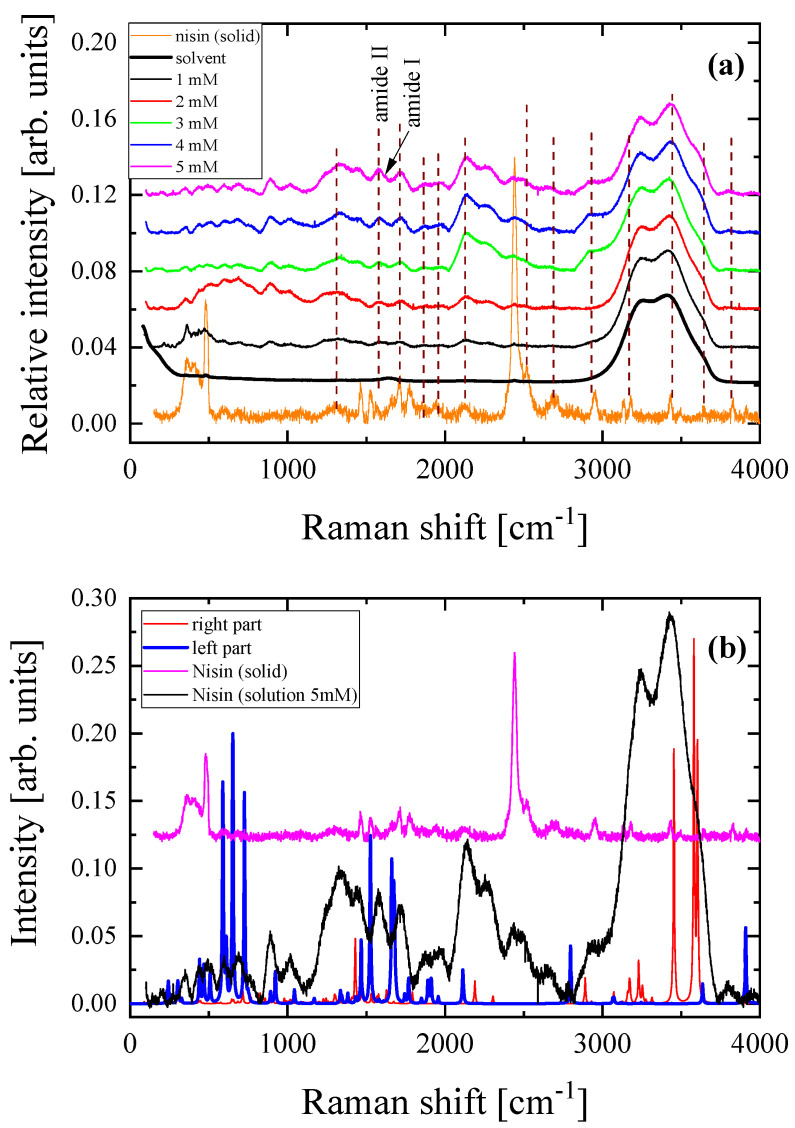
(**a**) Concentration dependence of the polarized (VV) Stokes-side Raman spectra of aqueous Nisin solutions in the dilute region corresponding to concentrations from 0 (solvent) to 5 mM with steps of 1 mM. The spectrum of Nisin in the solid state is also shown at the bottom. (**b**) Theoretically predicted Raman spectra corresponding to the binding regions of the two monomers constituting the dimer denoted by ellipses in Figure 2b. See text for details. The experimental Raman spectra of pure Nisin in the solid state and in solution at the concentration of 5 mM are also shown for comparison.

**Figure 4 antibiotics-12-00221-f004:**
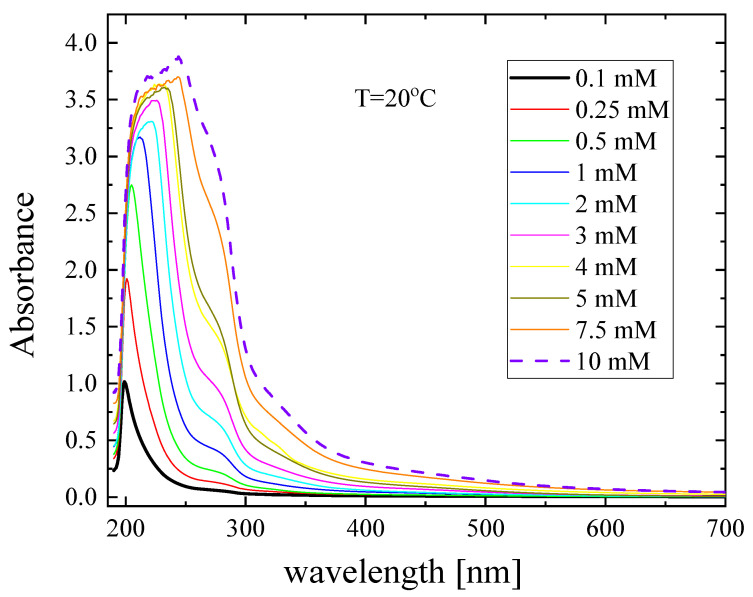
UV-Vis absorption spectra of Nisin as a function of concentration at 20 °C.

**Figure 5 antibiotics-12-00221-f005:**
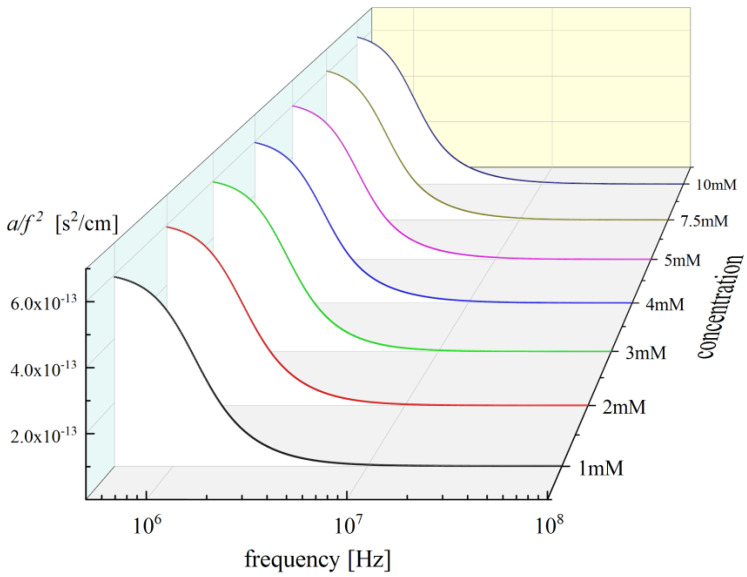
Acoustic absorption as a function of frequency at 20 °C in Nisin solutions. Solid lines represent the total fitting curves as received from the fitting procedure described in the text.

**Figure 6 antibiotics-12-00221-f006:**
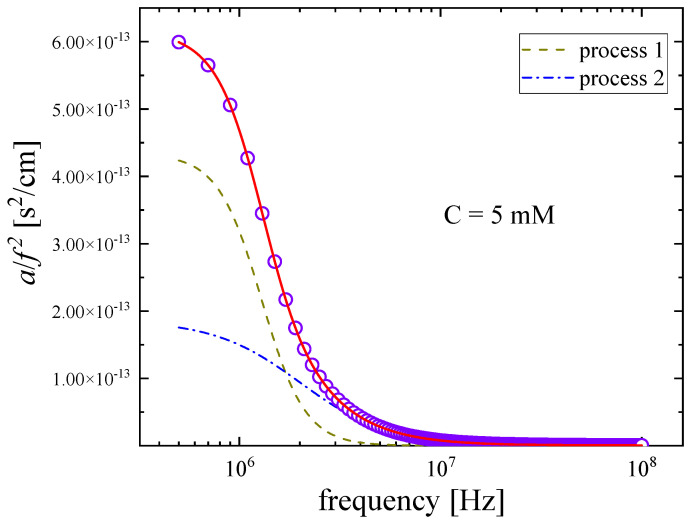
Representative fitting example of excess ultrasound absorption in a Nisin solution with C = 5 mM at 20 °C. The continuous red line represents the total relaxation curve, while the two individual dashed, and dot-dashed relaxation curves are assigned to distinct relaxation processes as described in the text. Circles correspond to the experimental data. Absorption coefficient α*/f*^2^ corresponding to the solvent was found to be frequency independent in the MHz range covered here and was subtracted from the absorption coefficient of the solution.

**Figure 7 antibiotics-12-00221-f007:**
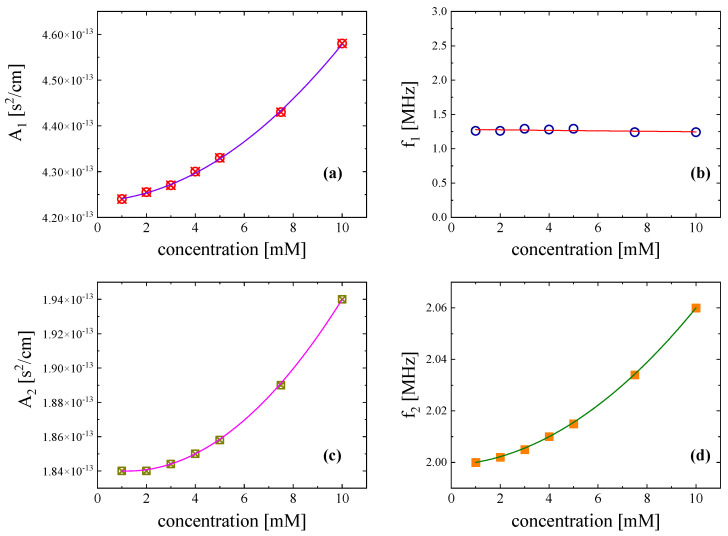
Ultrasonic relaxation amplitude *A_i_* (**a**,**c**) and relaxation frequency *f_ri_* (**b**,**d**) as functions of Nisin concentration at 20 °C. The error of relaxation frequency *f_r_* was estimated to be lower than ±2.5%, while the corresponding error for A and B was below ±5%. Subscripts 1 and 2 denote the acoustic parameters related to the conformational changes and self-aggregation of Nisin, respectively.

**Figure 8 antibiotics-12-00221-f008:**
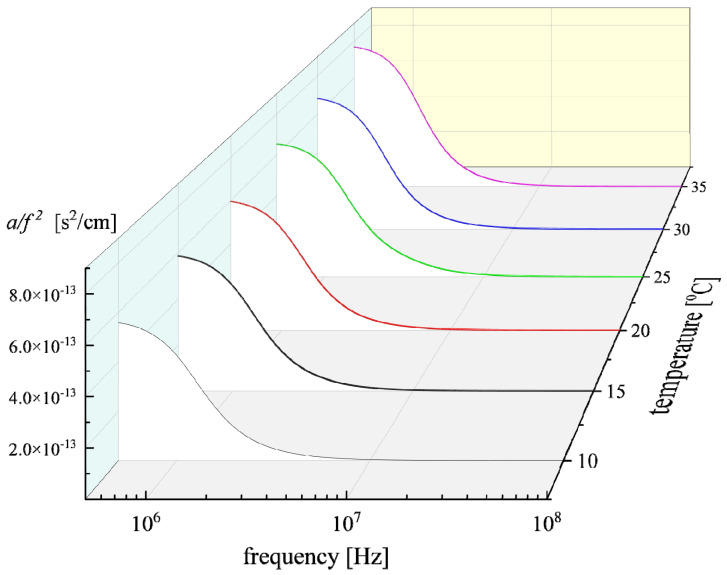
Frequency-reduced ultrasonic absorption *α*/*f*^2^ for Nisin solution at concentration C = 3 mM. Continuous lines represent relaxation curves for each temperature after the fitting procedure. Reduction in the *α*/*f*^2^ ratio with the increase in frequency was observed for all temperatures, which is typical of an ultrasonic relaxation process.

**Figure 9 antibiotics-12-00221-f009:**
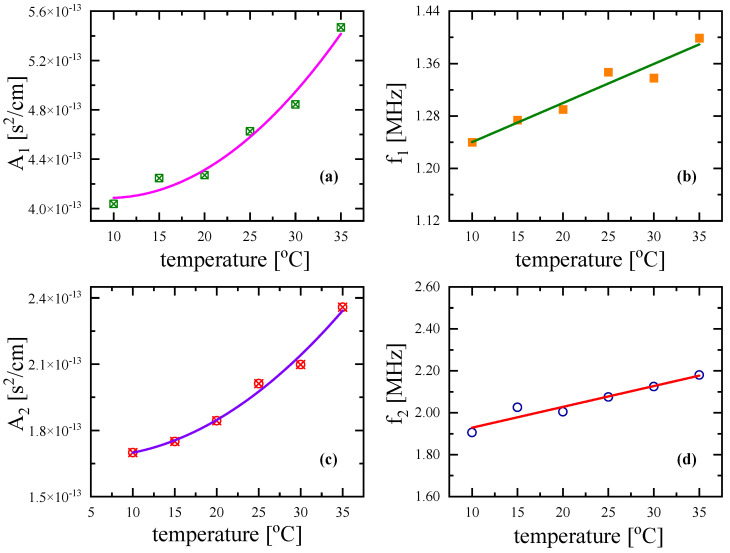
Ultrasonic relaxation amplitude *A_i_* (**a**,**c**) and relaxation frequency *f_ri_* (**b**,**d**) as functions of temperature for Nisin solution at concentration C = 3 mM. As in Figure 7, subscripts 1 and 2 denote the acoustic parameters related to the conformational changes and self-aggregation of Nisin, respectively.

**Figure 10 antibiotics-12-00221-f010:**
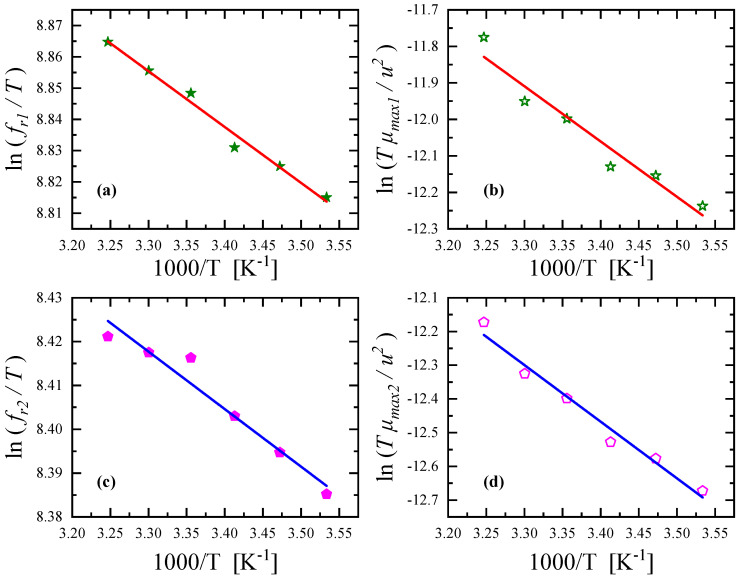
Relaxation frequencies as functions of reciprocal temperature for conformational change (**a**) and aggregation mechanism (**c**). Graphs of ln(*Tμ_max_/u*^2^) versus 1*/T* for conformational change (**b**) and aggregation mechanism (**d**).

**Figure 11 antibiotics-12-00221-f011:**
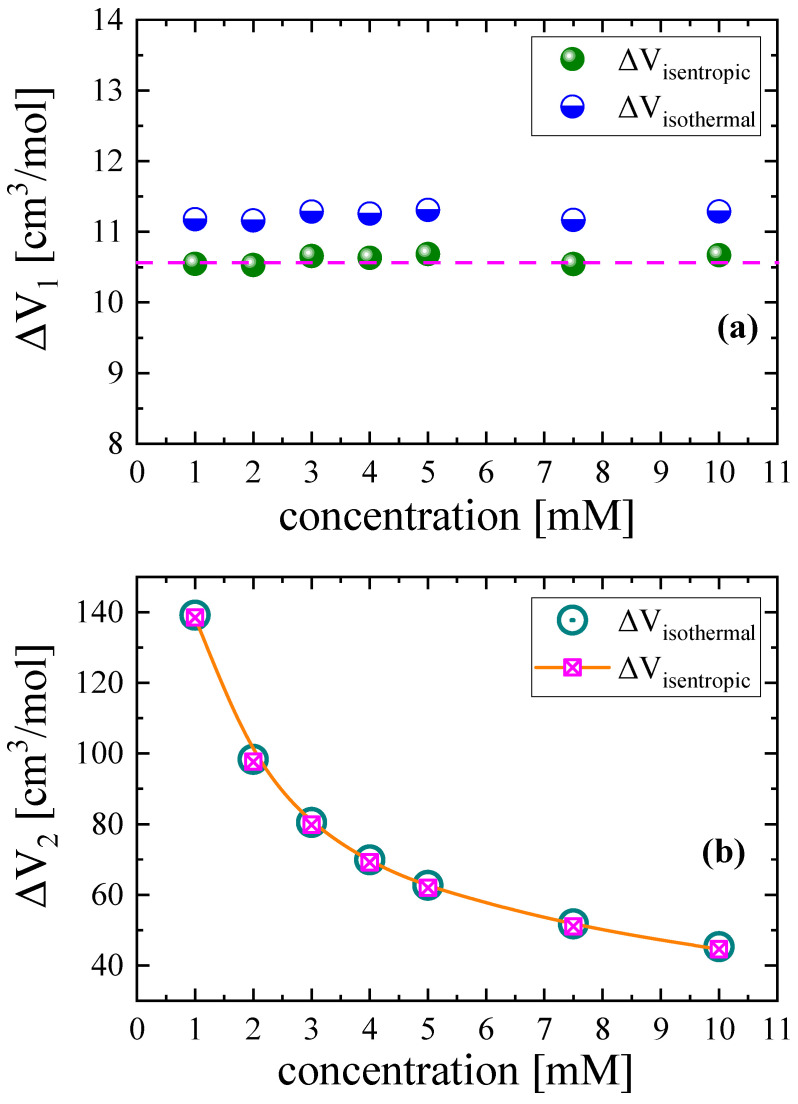
Isothermal and isentropic volume changes due to confirmational change (**a**) and self-aggregation (**b**) as functions of Nisin concentration. The enthalpy term contributes only slightly, and isothermal and isentropic volume changes are comparable in both relaxation processes.

**Figure 12 antibiotics-12-00221-f012:**
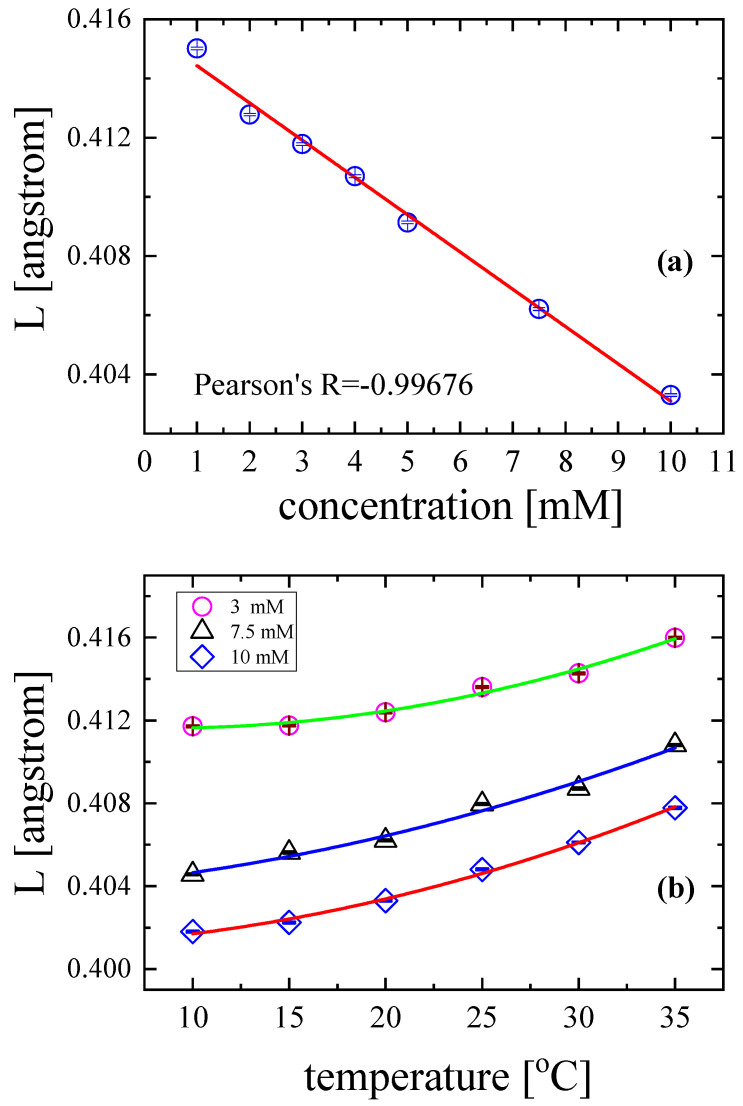
Intermolecular free length as a function of concentration at 20 °C (**a**) and temperature (**b**) at three selected solution concentrations.

**Table 1 antibiotics-12-00221-t001:** Properties of Nisin solution at concentration C = 3 mM at all temperatures studied.

*T* (°C)	*u* (m/s)	*ρ* (g/cm^3^)	*A*_1_ (×10^−13^ s^2^/cm)	*B*_1_ (×10^−16^ s^2^/cm)	*f_r_*_1_ (MHz)	*A*_2_ (×10^−13^ s^2^/cm)	*B*_2_ (×10^−16^ s^2^/cm)	*f_r_*_2_ (MHz)	*μ_max_* _1_	*μ_max_* _2_	*r* _1_	*r* _2_
10	1462.0	1.0069	4.04	1.46	1.24	1.70	3.80	1.91	0.0366	0.0237	0.0232	0.0150
15	1479.3	1.0062	4.25	1.46	1.27	1.75	4.42	2.03	0.0400	0.0262	0.0253	0.0166
20	1494.8	1.0051	4.27	1.46	1.29	1.84	3.26	2.00	0.0412	0.0276	0.0260	0.0175
25	1508.3	1.0037	4.63	1.46	1.35	2.01	4.37	2.08	0.0470	0.0315	0.0297	0.0199
30	1521.3	1.0024	4.84	1.46	1.34	2.10	3.68	2.12	0.0493	0.0339	0.0311	0.0215
35	1531.4	0.9999	5.47	1.46	1.40	2.36	3.28	2.18	0.0586	0.0394	0.0369	0.0249

**Table 2 antibiotics-12-00221-t002:** Properties of Nisin solutions at all temperatures studied at 20 °C.

*C* (mM)	*u* (m/s)	*ρ* (g/cm^3^)	*A*_1_ (×10^−13^ s^2^/cm)	*B*_1_ (×10^−16^ s^2^/cm)	*f_r_*_1_ (MHz)	*A*_2_ (×10^−13^ s^2^/cm)	*B*_2_ (×10^−16^ s^2^/cm)	*f_r_*_2_ (MHz)	*μ_max_* _1_	*μ_max_* _2_	*r* _1_	*r* _2_
1	1488.6	1.0007	4.24	1.46	1.26	1.84	2.55	2.000	0.0398	0.0274	0.0252	0.0174
2	1494.8	1.0032	4.26	1.45	1.26	1.84	2.48	2.002	0.0401	0.0275	0.0254	0.0174
3	1497.0	1.0051	4.27	1.46	1.29	1.84	2.26	2.005	0.0412	0.0276	0.0261	0.018
4	1499.2	1.0074	4.30	1.47	1.28	1.85	2.54	2.010	0.0413	0.0279	0.0261	0.0177
5	1503.5	1.0093	4.33	1.46	1.29	1.86	2.26	2.015	0.0420	0.0282	0.0266	0.0179
7.5	1510.6	1.0144	4.43	1.46	1.24	1.89	2.46	2.034	0.0415	0.0290	0.0262	0.0184
10	1517.4	1.0198	4.58	1.45	1.24	1.94	2.46	2.060	0.0431	0.0303	0.0272	0.0192

**Table 3 antibiotics-12-00221-t003:** The mass of solid Nisin from *Lactococcus lactis* 2.5% (balance sodium chloride) weighed for the preparation of the solutions.

Concentration in mM of Νisin Solutions	Mass in g of Solid Nisin from *Lactococcus lactis* 2.5% (Balance Sodium Chloride)
1	0.0168
2	0.0335
3	0.0503
4	0.0671
5	0.0839
7.5	0.1258
10	0.1677

## Data Availability

Data are available upon request from the corresponding author.

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
