# Peer review of "Evidence of Self-Association and Conformational Change in Nisin Antimicrobial Polypeptide Solutions: A Combined Raman and Ultrasonic Relaxation Spectroscopic and Theoretical Study"

_antibiotics, 2023, doi:10.3390/antibiotics12020221_

Round 1

Reviewer 1 Report

This manuscript entitled with "Evidence of Self-association and Conformational Change in Nisin antimicrobial polypeptide solu-tions: A combined Ra-man and Ultrasonic Relaxation spectroscopic and theoretical study" conducted a systematic analysis of molecular relaxation dynamics of aqueous Nisin solutions. They showed good aggrement of prediction with experimental results. This manuscript was well thought and but not orgnized regirously. It can be accepted for publication if the following concerns can be addressed:

  1. Abstract should mention the objective and significance briefly.
  2. Discussion of the results should include the comparison with peers' work.
  3. In the main text, some places write as "Fig. #" while some place write as "Figure #". Please keep consistent.
  4. Fig. 3(b) was not described and discussed in the main text.
  5. Line numbers and page numbers were messed.

Author Response

  • Reviewer #1 comments:

This manuscript entitled with "Evidence of Self-association and Conformational Change in Nisin antimicrobial polypeptide solutions: A combined Ra-man and Ultrasonic Relaxation spectroscopic and theoretical study" conducted a systematic analysis of molecular relaxation dynamics of aqueous Nisin solutions. They showed good agreement of prediction with experimental results. This manuscript was well thought and but not organized rigorously. It can be accepted for publication if the following concerns can be addressed:

Issue 1:

Abstract should mention the objective and significance briefly.

Reply to Reviewer comment and changes made:

Nisin polypeptide is characterized by antibacterial properties, making it a compound with many applications, mainly in the food industry. As a result, a deeper understanding of its behavior, especially after its dissolution in water, is of the utmost importance. This could be possible through the study of aqueous solutions of Νisin by combining vibrational and acoustic spectroscopy techniques.

This comment is added in the revised manuscript and is highlighted with yellow color (lines: 14-18).

Issue 2:

Discussion of the results should include the comparison with peers' work.

Reply to Reviewer comment and changes made:

The relevant research activity is limited for this compound and thus there is not much possibility to compare our results with earlier research works. However, we presented information that strengthens our hypothesis of self-aggregation of nisin. The possibility of formation of dimeric and tetrameric structures of nisin has been reported in the literature [19].

This comment is added in the revised manuscript and is highlighted with yellow color. Α new reference was also added, and the reference list is renumbered, accordingly (lines: 193-195).

Issue 3:

In the main text, some places write as "Fig. #" while some place write as "Figure #". Please keep consistent.

Reply to Reviewer comment and changes made:

We agree with reviewer, and we replaced all the expressions “Fig. #” with “Figure #” in several parts of the manuscript. All changes are highlighted with yellow color.

Issue 4:

Fig. 3(b) was not described and discussed in the main text.

Reply to Reviewer comment and changes made:

We added and highlighted the comment “as shown in Figure 3(b)” to show where we described further Figure 3(b) (line: 212-213).

Peaks observed in the theoretically calculated vibrational spectra for the two parts of dimer were also present in the experimental spectrum of the aqueous solution of Nisin mainly in the frequency range 1500 - 1700cm-1.

This note added in the revised manuscript and highlighted with yellow color (lines: 215-217).

Issue 5:

Line numbers and page numbers were messed

Reply to Reviewer comment and changes made:

We agree with reviewer, and we modified the numbering of lines and pages.

Reviewer 2 Report

1. In the following below line of Introduction section, an-tibiotic re-sistance should be written as antibiotic resistance & anti-biotic peptides as antibiotic  peptides.

The World Health Organization aiming to prevent an-tibiotic re-sistance recognises the development of new antibacterial compounds and more specifically anti-biotic peptides, as promising materials for this purpose .

Please incorporate such type of changes and remove grammatical errors throughout the manuscript.

2. Remove grammatical errors such as com-pounds to compounds, bacte-rium to bacterium, vi-brational to vibrations.

3. Italicize species name such as Lactococcus lactis throughout the manuscript.

4. Mention PDB ID of Nisin monomer obtained from RCSB database.

5. In Fig.2 b label the interacting residues and label th distance between the bonds of interacting residues. Improve the resoultion of the figure also.

6. Conclusion lacks a strong rationale for the objective of this study. 

7. Several grammatical errors and incomprehensible sentances are in the manuscript. Improve the sentances and make necessaru corrections throught the manuscript.

Author Response

  • Reviewer #2 comments:

Comments and Suggestions for Authors

Issue 1:

In the following below line of Introduction section, an-tibiotic re-sistance should be written as antibiotic resistance & anti-biotic peptides as antibiotic  peptides.

The World Health Organization aiming to prevent an-tibiotic re-sistance recognises the development of new antibacterial compounds and more specifically anti-biotic peptides, as promising materials for this purpose .

Please incorporate such type of changes and remove grammatical errors throughout the manuscript.

Reply to Reviewer comment and changes made:

We agree with reviewer, and we corrected grammatical errors in several parts of the manuscript. All changes were highlighted with green color in the revised manuscript.

Issue 2:

Remove grammatical errors such as com-pounds to compounds, bacte-rium to bacterium, vi-brational to vibrations.

Reply to Reviewer comment and changes made:

We agree with reviewer, and we removed grammatical errors throughout the manuscript. All changes were highlighted with green color (lines: 50, 52, 67).

Issue 3:

Italicize species name such as Lactococcus lactis throughout the manuscript.

Reply to Reviewer comment and changes made:

We agree with reviewer, and we italicized species names. All changes are highlighted with green color in the revised manuscript (lines: 52, 54, 56, 90, 98 and Table 1).

Issue 4:

Mention PDB ID of Nisin monomer obtained from RCSB database.

Reply to Reviewer comment and changes made:

The PDB ID of Nisin monomer obtained from RCSB database is 1WCO. We added and highlighted with green color the PDB ID in revised manuscript (line: 124).

Issue 5:

In Fig.2 b label the interacting residues and label the distance between the bonds of interacting residues. Improve the resolution of the figure also.

Reply to Reviewer comment and changes made:

We agree with reviewer, and we modified Figure 2(b) in order to distinguish the two interacting monomers and add the length of the hydrogen bonds. The resolution is also improved.

Issue 6:

Conclusion lacks a strong rationale for the objective of this study.

Reply to Reviewer comment and changes made:

Tetramer formation is not favored based on theoretical calculations, although their existence has been reported in the literature (lines: 193-195).

Τhe assumption that conformational change is the first mechanism is confirmed by the fact that the relaxation frequency does not change significantly in the studied concentration range. Raman and UV-Vis spectroscopy results support the existence of self-aggregation phenomenon in nisin aqueous solutions.

We revised the conclusions part according to reviewer’s suggestions. All changes are highlighted with green color (lines: 492-493, 500-504).

Issue 7:

Several grammatical errors and incomprehensible sentances are in the manuscript. Improve the sentances and make necessaru corrections throught the manuscript.

Reply to Reviewer comment and changes made:

We agree with reviewer’s comment, and we revised the English language of the manuscript in several parts.

Round 2

Reviewer 2 Report

Improve the conclusion section with elucidating the significance and citing strong rationale for the study.

Author Response

Reviewer #2 comments:

Issue 1:

Improve the conclusion section with elucidating the significance and citing strong rationale for the study.

Reply to Reviewer comment and changes made:

The conclusions section is revised according to reviewer’s suggestions in an effort to elucidate the significance of the present study.

All changes added in the revised manuscript, are highlighted with yellow color (lines: 517-522 and 525-526).
